# Comparative Analysis of Proteomes of a Number of Nosocomial Pathogens by KEGG Modules and KEGG Pathways

**DOI:** 10.3390/ijms21217839

**Published:** 2020-10-22

**Authors:** Mikhail V. Slizen, Oxana V. Galzitskaya

**Affiliations:** 1Institute of Protein Research, Russian Academy of Sciences, 142290 Pushchino, Moscow Region, Russia; mikha.shtol@gmail.com; 2Institute of Theoretical and Experimental Biophysics, Russian Academy of Sciences, 142290 Pushchino, Moscow Region, Russia

**Keywords:** KEGG modules, KEGG pathways, drug resistance, *Staphylococcus aureus*, *Enterobacter* spp., *Pseudomonas aeruginosa*, *Mycoplasma* spp.

## Abstract

Nosocomial (hospital-acquired) infections remain a serious challenge for health systems. The reason for this lies not only in the local imperfection of medical practices and protocols. The frequency of infection with antibiotic-resistant strains of bacteria is growing every year, both in developed and developing countries. In this work, a pangenome and comparative analysis of 201 genomes of *Staphylococcus aureus*, *Enterobacter* spp., *Pseudomonas aeruginosa*, and *Mycoplasma* spp. was performed on the basis of high-level functional annotations—KEGG pathways and KEGG modules. The first three organisms are serious nosocomial pathogens, often exhibiting multidrug resistance. Analysis of KEGG modules revealed methicillin resistance in 25% of *S. aureus* strains and resistance to carbapenems in 21% of *Enterobacter* spp. strains. *P. aeruginosa* has a wide range of unique efflux systems. One hundred percent of the analyzed strains have at least two drug resistance systems, and 75% of the strains have seven. Each of the organisms has a characteristic set of metabolic features, whose impact on drug resistance can be considered in future studies. Comparing the genomes of nosocomial pathogens with each other and with *Mycoplasma* genomes can expand our understanding of the versatility of certain metabolic features and mechanisms of drug resistance.

## 1. Introduction

A nosocomial infection (hospital-acquired infections, HAI) is an infection acquired as the result of hospitalization or visiting a medical facility [1]. Nosocomial infections have long been a major problem for health care system. Millions of patients worldwide suffer from hospital-acquired infections every year, resulting in significant deaths and financial losses for health systems. Between 7% (in developed countries) and 10% of patients (in developing countries) acquire at least one nosocomial infection during hospitalization (WHO, 2018). The main types of nosocomial infections include hospital-acquired pneumonia associated with artificial ventilation, catheter-associated bloodstream infections, catheter-associated urinary tract infections, and surgical infections [2].

A particular problem of nosocomial infections is widespread drug-resistant strains that significantly complicate the treatment [3]. Antibiotic resistance has spread rapidly around the world in recent years and poses a critical threat to public health. Antibiotic resistance genes are spread both within individual species of organisms and horizontally between different species and strains. In total, about 12 species of bacteria are responsible for most of HAI cases, including those caused by methicillin-resistant strains of *Staphylococcus aureus* (*S. aureus*), *Pseudomonas aeruginosa* (*P. aeruginosa*) and *Enterobacter* spp. [4].

*S. aureus* is a Gram-positive bacterium, a widespread inhabitant of the skin and upper respiratory tract of humans. Various strains of *S. aureus* cause a wide range of hospital infections. Methicillin-resistant *S. aureus* (MRSA) strains are the most common cause of nosocomial infections (HA-MRSA) [5]. In addition to hospital strains, community-associated strains (CA-MRSA) and livestock-associated strains (LA-MRSA) are of widespread occurrence. Previously, CA-MRSA strains were associated with skin and soft tissue infections, just as HA-MRSA strains were associated with severe pneumonia and blood infections [6]. However, this distinction between CA-, LA-, and HA-MRSA strains has become less clear in recent years, since blood infections caused by LA- and CA-MRSA strains are increasingly common in hospitals [7]. Cases of multidrug resistance, such as penicillin resistance, of MRSA strains are increasingly frequently reported. Overall, MRSA infections impose an additional burden on health care, as they increase hospital stays, health care costs, and reduce quality of life. The death rate from *S. aureus* bloodstream infection exceeds 20% [8].

*P. aeruginosa* is a widespread Gram-negative bacterium, a human opportunistic microorganism that may cause severe respiratory infections in immunocompromised patients. *P. aeruginosa* causes 10% of all nosocomial infections, however, cases of community-acquired infections caused by this microorganism are becoming more frequent. The plasticity and adaptability of the *P. aeruginosa* genome, provided by a large number of regulatory genes (about 8% of the genome), allows the pathogen to persist for a long time in the host organism and resist antibiotic treatment [9]. Originally resistant to a wide range of antimicrobial agents, *P. aeruginosa* is currently resistant to several classes of antibiotics [10]. With a high ability to acquire and preserve antibiotic resistance genes from other organisms, *P. aeruginosa* strains remain the cause of nosocomial infections [11]. Moreover, there are frequent reports about the resistance of *P. aeruginosa* clinical isolates to “last resort” antibiotics from the classes of polymixins and carbapenems [12]. *P. aeruginosa* colonizes humid environments and can often be found in wounds, ventilators and oxygenator tubes, and urinary catheters, where biofilm formation ensures the microorganism’s survival, its evasion of immunity and resistance to various antimicrobial drugs [13].

*Enterobacter* is a genus of Gram-negative rod-shaped bacteria, some of which are part of the normal human intestinal biome. Various *Enterobacter* strains can cause infections associated with the insertion of catheters (e.g., urethral infections), upper respiratory tract diseases, and skin diseases. For many years, the most dangerous representatives of the genus were *Enterobacter aerogenes* (recently renamed *Klebsiella aerogenes*) and *Enterobacter cloacae*, which are a problem in neonatal and intensive care units, especially for patients with artificial ventilation [14]. Multidrug-resistant *Enterobacter* strains are an increasing threat. For example, the antibiotic-resistant strains of *E. aerogenes* ST4 and ST93 are responsible for an increasing share of nosocomial infections in the United States [15]. Carbapenem-resistant *E. cloacae* ST178 and ST78 strains are spreading to medical facilities throughout the United States [16]. Until 2005, 99.3% of all *Enterobacter* spp. clinical isolates were sensitive to carbapenems [17]. However, at the moment, according to the WHO, resistance to carbapenems is documented in all observed regions of the planet. Moreover, strains of *E. aerogenes* with full drug resistance, insensitive even to colistin, the antibiotic of “last resort”, have already been discovered [18]. The situation is complicated by the ability of *E. aerogenes* to harbor subpopulations of colistin-resistant bacteria, which makes diagnosis with modern methods more difficult [19].

The problem of antibiotic resistance is compounded by the possibility of mixed infections. Mixed bacterial cultures can form stable multi-species communities, such as biofilms. These communities provide additional benefits to pathogens, including metabolic cooperation, where one species uses a metabolite produced by other species, and increased resistance to antibiotics or host immune responses compared to single-species biofilms [20].

The identification of various related organisms’ genomes laid the foundation for comparative analysis methods that raised new questions [21]. As comparative genomics developed, the term *pangenome* was introduced, referring to the entire set of genes for all strains within a bacterial species. Initially, the pangenome was divided into two parts: the core genome, which represents the set of genes common to all strains, and the dispensable genome, containing strain-specific genes or genes common to a subset of strains. However, it is now more common to divide the pangenome into three main parts: the core genome, the dispensable or accessory genome, and the (unique) genome.

The core genome forms the basis of the species phylogeny and is considered to be representative at various taxonomic levels [22]. The accessory genome can encode additional biochemical pathways that provide some benefits, such as adaptation to the environment, specific virulence, or antibiotic resistance [23]. With a set of genomes of the same species strains, it is possible to study the evolution of a species by discovering its genomic diversity with the use of comparative analysis. Core genes shared by all pathogenic species strains can be used as potential targets for antibiotics [24]. Pangenome analysis is particularly useful in the study of pathogenic strains. Thus, the large-scale analysis of *P. aeruginosa* [25] showed the high genomic variability in the species strains, including genes encoding proteins, which were used as targets for developing vaccines, which partially explains high resistance of the bacteria to vaccination and other therapies. In the pangenome study of *Acinetobacter baumannii* [26], the conserved regions of the genome were analyzed at the protein level, resulting in the discovery of new potential targets for vaccine design.

## 2. Results and Discussion

### 2.1. Comparison of Single Genomes of S. aureus, E. cloacae, P. aeruginosa, M. mobile

The number of genes, KEGG modules, and KEGG pathways for the corresponding strains in the KEGG database are presented in Table 1.

The comparative results of the modules’ occurrence in the organisms are shown in Figure 1A and Table 2. Theoretically, the modules could be divided into 15 categories, but six categories were empty. Within each category, the modules were divided into five subcategories: modules related to antibiotic resistance, metabolic modules, transport modules, two-component regulatory modules, and modules related to antimicrobial peptide (AMP) resistance. The division was largely arbitrary. For example, many resistance modules are efflux systems, and some transport modules can be considered as protection modules against external AMPs, as well as modules for the transport of endogenous AMPs. The comparison of the modules’ occurrence resulted in a number of facts, which were in a good agreement with the ecological niche of organisms and with genome size.

For *S. aureus*, 38 unique modules belonging to all five categories were found (Table 2). Among them, there are 10 modules of resistance, including three modules of resistance to antimicrobial peptides. Four modules of resistance—M00700 (multidrug resistance, efflux pump AbcA), M00702 (multidrug resistance, efflux pump NorB), M00704 (tetracycline resistance, efflux pump Tet38), and M00705 (multidrug resistance, efflux pump MepA)—belong to antibiotic efflux systems. It is interesting to note that *S. aureus* has modules of resistance to bacitracin (M00737) and lantibiotics (M00817), as well as two modules related to the transport and regulation of the transport of antimicrobial peptides (M00732 and M00733, respectively). This is consistent with the numerous reports of *S. aureus* multidrug resistance. Even non-pathogenic strains are involved in a chemical “war” with Gram-negative and Gram-positive bacteria, which partly explains this phenomenon. The abundance of mobile elements (transposons, plasmids, phages, etc.) in the *S. aureus* genome provides an easy transfer of drug resistance.

A variety of both unique and common metabolic and transport modules for *P. aeruginosa* and *E*. *cloacae* was found. This fact is explained by the wide range of habitats and substrates for *P. aeruginosa* and the complex environment of the intestinal biome for *E. cloacae*.

For *P. aeruginosa*, 48 unique modules were found. Among them, there are modules related to the quorum sensing and biofilm formation (M00820) and access and transport of iron (M00259, M00328), which correspond to the characteristics of bacteria described in the literature. Eight of the detected modules (seven modules of the Mex and TriABC-TolC efflux systems) are related to the efflux of antibiotics, which correspond to the broad antibiotic resistance of the microorganism. In addition, the imipenem resistance M00745 module (repression of porin OprD) was found to be unique for this species. It is noteworthy that this species of bacteria was isolated from the surface of an onion, but not from a clinical sample. Thus, it can be assumed that these gene modules are widespread not only in clinical but also in natural conditions.

Seventy-five unique modules were found for *E. cloacae*. Among them, there are the M00060 module related to the lipopolysaccharide component synthesis, five multidrug resistance modules (efflux systems AcrAD-TolC, MdtABC, MdtABC, and OqxAB, and the OqxAB transporter), and the glycogen breakdown module (M00855). It is noteworthy that there are four modules of resistance to antimicrobial peptides related to membrane modification.

A total of 65 modules were found to be common to both *E. cloacae* and *P. aeruginosa*. Most of them are related to the metabolism and transport of substances. It is noteworthy that there were five common modules of antibiotic resistance: M00647 (AcrAB-TolC/SmeDEF efflux system), the macrolide resistance module M00709 MacAB-TolC transporter, M00711 MdtIJ efflux system, and two macrolide resistance modules, M00742 and M00743, corresponding to FtsH and HtpX proteases.

Nine out of 12 common modules, exclusively for *S. aureus* and *E. cloacae*, were involved in the assimilation of nutrients, primarily sugars.

*E. cloacae* and *M. mobile* had only one common module, the M00854 glycogen synthesis module.

In the SEP group (annotations that only *M. mobile* lacks), 39 modules were annotated. The absence of many metabolic modules in *M. mobile*, including the synthesis of some amino acids, fatty acids, and sugars, was confirmed. This was entirely consistent with the ecological role of the organism—highly specialized cellular parasitism.

Only 13 common modules were annotated for each of the analyzed organisms, some of them absolutely essential: M00002 is the main module of glycolysis, M00005 is the module of ribose-5P synthesis, M00120 is the module of coenzyme A synthesis, M00157 is the module of F-type ATPase, M00178 is the module of the bacterial ribosome, M00201 is a module of the alpha-glucoside transport system, M00222 is a phosphate transport system module, M00260 is the module of the DNA polymerase III complex, M00273 is module of the phosphotransferase system (PTS), fructose-specific II component, M00299 is the module of the spermidine/putrescine transport system, M00307 is the pyruvate oxidation module, M00360 is the module of aminoacyl-tRNA synthesis in prokaryotes, and M00579 is the module of the phosphate acetyltransferase–acetate kinase pathway.

It can be noted that each analyzed organism is characterized by a certain set of PTSs associated with the active transport of simple sugars, which results in the ability of the organisms to absorb various sugars.

Since the modules contain complete groups of genes that are necessary for the implementation of a function, an additional comparison for the presence of KEGG pathways was carried out. A pathway can be ascribed to an organism even if it does not contain the full set of necessary proteins. Thus, it is possible to detect certain properties of the organism, even if some of the proteins required to realize this property are not detected. The comparison results are shown in Figure 1B.

About 80% of all pathways were found either in all organisms, or in all but *M. mobile*, a highly specialized parasite with a reduced genome.

The remaining pathways are mainly associated with the degradation of specific substances and metabolic nuances (with the exception of *S. aureus*, which has two pathways related to infection).

It is noticeable that for *M. mobile*, the number of KEGG pathways significantly exceeds the number of KEGG modules. On the one hand, this can be explained by the inclusion of some pathways that do not function in the organism, but their proteins are involved in the work of other, complete pathways. On the other hand, some metabolic pathways of the organism can be supplemented by pathways of the host cell, which makes it possible to get rid of the links necessary in other conditions.

### 2.2. Analysis of Pangenomes and Core Genomes

#### 2.2.1. *S. aureus*

Complete lists of KEGG modules and KEGG pathways for the *S. aureus* pangenome are given in Appendix A
Appendix A and Figure 2.

A total of 53 different modules were annotated in the *S. aureus* pangenome. Twenty-six of them (49%) were annotated for all analyzed strains (core modules). Twenty of them were modules of basic metabolism, such as modules of glycolysis, the pentose phosphate pathway, the first oxidation of the Krebs cycle, the synthesis of coenzyme A, etc. Two modules were related to the synthesis of staphyloferrins, *Staphylococcus*-specific siderophores. It is noteworthy that the core genome contained two modules of multidrug resistance—the efflux systems AbcA and NorB. The ABC efflux system is common among all organisms and provides exotoxin elimination. Blocking this system significantly increases the effectiveness of antibiotics, which makes it a promising universal target for increasing the effectiveness of therapies [27]. The Nor efflux systems are unique to staphylococci and provide resistance to quinolones and other antibiotics. It is also noteworthy that the core genome contains two modules of resistance to cationic antimicrobial peptides (M00726 and M00730). The protection mechanism is based on the compensation of bacterial membranes’ negative charge by modifying the membrane components with cations, such as lysine residues. The prevalence of such systems among *S. aureus* strains complicates the use of existing and promising therapies with antimicrobial peptides.

Among the annotations not related to the core genome, modules of drug resistance are worth noting. Thus, the efflux systems of tetracyclines, Et38, and several antibiotics, MepA, and the dltABCD operon that increases resistance to antimicrobial peptides, were found in 45 of the 53 analyzed strains. The beta-lactam resistance module (Bla) was found in 24 genomes, the methicillin resistance module in 13 genomes, and the QacA efflux system multidrug resistance module in 11 genomes. Thus, most of the known staphylococcal strains, in addition to methicillin resistance, have resistance to many vital drugs.

Of the 108 pathways annotated in the pangenome, 105 belong to the core genome. It is noteworthy that all modules of amino acid biosynthesis (e.g., isoleucine), annotated only in some of the strains, have annotations of the corresponding pathways in the core genome. It is noteworthy that the modules of amino acid biosynthesis (e.g., isoleucine), were annotated only in some of the strains, while the pathways corresponding to them have annotations in the core genome. This indicates that at least some elements of these metabolic pathways are inherent characteristics in each *S. aureus* strain. The pathway of bacterial invasion of epithelial cells was annotated only in 37 out of 53 genomes. The degradation pathways of aminobenzoate and styrene were annotated in one genome and two genomes, respectively.

#### 2.2.2. *Enterobacter* spp.

Complete lists of KEGG modules and KEGG pathways for the *Enterobacter* spp. pangenome are given in Appendix A and Figure 3.

For the *Enterobacter* spp. Pangenome, 103 modules were annotated. These include 34 modules annotated for all 35 strains (core modules), 23 modules annotated for 34 strains, and 14 for 33 strains, which is 69% ((34 + 23 + 14)/103) of all modules in the pangenome. It is noteworthy that the core genome contains modules of bacterial “glycogen” synthesis and degradation. The synthesis of the glycogen-like polymer of glucose by some bacteria is likely to occur as a response to stress and is intended to accumulate nutrients, but this is not true for every bacterium with this ability [28]. M00063 (CMP-KDO biosynthesis) and M00064 (ADP-L-glycero-D-manno-heptose biosynthesis) modules were annotated for 33 of 35 strains, and the M00060 module (KDO2-lipid A biosynthesis, Raetz pathway, LpxL-LpxM type) associated with the synthesis of bacterial lipopolysaccharide components was annotated for 32 of 35 strains. Twenty-six out of 35 strains have the AcrEF-TolC efflux system module. The system, originally described in *E. coli*, provides bacterial resistance to quinolones and tigecycline. The carbapenem resistance module (M00851) was found in 11 strains. This is a small module consisting of just one protein—any of the 21 beta-lactamase variants. Only one strain has an annotated M00746 module, which is responsible for the repression of porin OmpF. The three-subunit porin OmpF provides passive transport of substances into the cells of Gram-negative bacteria. Its adaptive repression greatly increases the resistance of bacteria to a wide range of substances.

Thus, at least one multidrug resistance module was found in 76% of the analyzed strains. Carbapenem resistance was found in 31% of the strains.

Of the 125 pathways annotated for the *Enterobacter* spp. pangenome, 109 belong to the core genome. Most of the non-core genome pathways are responsible for the degradation of specific substances, such as dioxin, toluene, xylene, and atrazine. Such metabolic pathways have been described for soil bacterial communities, especially in areas of oil contamination.

#### 2.2.3. *P. aeruginosa*

Complete lists of KEGG modules and KEGG pathways for the *P. aeruginosa* pangenome are given in Appendix A and Figure 4.

A total of 80 modules were annotated in the *P. aeruginosa* pangenome, of which 64 related to the core genome. It is particularly remarkable that nine of the 80 modules provide antibiotic resistance. There are two such modules in the core genome: M00639 is the MexCD-OprJ efflux system and M00642 is the MexJK-OprM efflux system. Both modules provide *P. aeruginosa* with multidrug resistance. In addition, there is evidence that increased expression of these efflux systems attenuates the virulence processes associated with quorum sensing. In 20 of the 21 genomes, the M00718 module of the MexAB-OprM efflux system, which confers multidrug resistance, and the M00745 module of the OprD repression system, conferring resistance to imipenem, were found. The modules M00641 of the MexEF-OprN efflux system and M00769 of the MexPQ-OpmE efflux system were annotated in 17 genomes. The M00643 module of the MexXY-OprM efflux system was found in 16 genomes. The M00851 module of carbapenem resistance was found only in three genomes out of 21.

Thus, based on the pangenome data, *P. aeruginosa* has the widest spectrum of antibiotic resistance, but mainly due to a very developed efflux system. Systems of enzymatic cleavage of antibiotics were found in only 14% of strains. Of the 121 pathways annotated in the *P. aeruginosa* pangenome, 120 belong to the core genome, and only one pathway is carried out by 20 out of 21 strains (synthesis and cleavage of ketone bodies). The presence of xylene and toluene degradation pathways in the core genome is noteworthy, which means the presence of at least some pathway elements in all *P. aeruginosa* strains.

#### 2.2.4. *Mycoplasma* spp.

Complete lists of KEGG modules and KEGG pathways for the *Mycoplasma* spp. pangenome are given in Appendix A and Figure 5.

In total, remarkably few modules have been annotated in the *Mycoplasma* spp. pangenome—13—of which only one belongs to the core genome, M00005 module, which provides the synthesis of phosphoribosyl pyrophosphate from ribose-5-phosphate. In 91 of 93 genomes, the M00157 module of the F-type prokaryotic ATPase was annotated, and in 89, the main glycolysis M00002 module, associated with the metabolism of tricarboxylic substances. The rest of the modules are less universal, and all of them are associated with the basic metabolic functions, for example, the synthesis of acetyl-CoA from pyruvate (M00307 module) and others. It is noteworthy that one of the genomes contains a glycogen synthesis module—M00854. No specific modules of antibiotic resistance in *Mycoplasma* spp. were annotated.

A total of 73 pathways have been annotated in the *Mycoplasma* spp. pangenome, of which only 31 belong to the core genome. It is particularly remarkable that many seemingly universal pathways are not universal for this taxon, for example, the pathway of tryptophan metabolism (in 26 genomes), lysine degradation (in 25 genomes), fatty acid degradation (in eight genomes), etc. This means that many strains do not even have single elements of these metabolic pathways. This fact is explained by the overspecialized parasitism of the genus representatives, when many metabolic schemes may be associated with the metabolism of the host cell. In addition, the possible elements of these pathways may differ significantly from those in other bacteria and have not yet been described. At the moment, the proportion of genes with unclear function in *Mycoplasma* spp. varies from 30–60%.

### 2.3. Comparison of Core Genomes and Pangenomes

Since the comparison of individual genomes does not provide complete information about the whole spectrum of possible annotations of organisms, a comparison of genome groups was performed. The results are given in Figure 6. According to the results of the core genome comparison, 11 modules were shown to be unique for *S. aureus*: two multidrug resistance modules (AbcA and NorB efflux systems), modules for the synthesis of staphyloferrins A and B, two modules for resistance to antimicrobial peptides, formaldehyde assimilation module, chorismate synthesis module, etc. According to the comparison of pangenomes in *S. aureus*, additional unique modules of resistance to beta-lactams and methicillin, three modules of efflux systems, were found. At the same time, the number of unique metabolic modules was smaller.

Comparison of pangenomes revealed 32 modules unique to *Enterobacter* spp., and comparison of core genomes—only nine. Among these nine, the presence of glycogen synthesis and degradation modules, the glyoxylate cycle, and some modules of the pentose phosphate pathway, are notable. Of the 32 modules unique to the *Enterobacter* spp. pangenome, most are related to the metabolism of certain compounds, among them the nitrogen fixation module, the heme synthesis module, and the biotin and methionine synthesis modules, and are of interest. It is noteworthy that the pangenome contains two unique multidrug resistance modules—the OmpF porin repression module and the AcrEF-TolC efflux system.

For *P. aeruginosa*, the number of unique modules in the core genome was 37, while only 15 were found in the pangenome. This remarkable difference can be explained by the fact that the studied genomes of *P. aeruginosa* are much more similar to each other than the genomes of other organisms involved in the analysis. Although *P. aeruginosa* has a large genome and a developed ability for horizontal gene transfer, it can be concluded that the organism has a relatively large and unchanged “nucleus” of metabolism. Both the core genome and the pangenome of *P. aeruginosa* are characterized by the presence of a large number of unique antibiotic resistance modules. There are seven such modules in the pangenome, six of them are efflux systems of the Mex family and one porin repression module, OprD. Only two out of 37 core genome modules provide drug resistance (Mex efflux systems). Most of the remaining 35 modules are associated with various metabolic processes; among them, the module of lipopolysaccharide (LPS) component synthesis M00063, module of denitrification M00529, the GABA synthesis M00135 module, and the cobalamin synthesis M0122 module, are notable.

There are 32 modules common exclusively to *P. aeruginosa* and *Enterobacter* spp. in pangenomes, and 14 such modules are in the core genomes (fat beta-oxidation module and 13 biosynthetic modules, eight are responsible for the synthesis of amino acids). It is noteworthy that both core genomes contain the M00176 module of assimilative sulfate reduction.

No unique modules for *Mycoplasma* spp. were found. The absence of modules common for all analyzed organisms in the core genome is noteworthy. There were nine such modules in the pangenomes—modules of ATP, GTP, CoA, acetyl-CoA, and ribose 5-F synthesis and the main glycolysis module (not annotated for all *Mycoplasma* spp.).

The SEP group (without *Mycoplasma* spp.) includes mostly various metabolic modules. It is noteworthy that the *Mycoplasma* spp. pangenome lacks the Krebs cycle and the synthesis of fatty acids, arginine, isoleucine, threonine, cysteine, tryptophan, and glutamate. This means that none of the analyzed strains of *Mycoplasma* spp. are able to synthesize these substances.

Below are the comparative results for pathways in the pangenomes of organisms (Figure 7).

The almost complete absence of unique pathways in the organisms is remarkable. For *S. aureus*, these are five pathways in the pangenome and six pathways in the core genome. These are specific pathways of staphylococcal infection, pathways related to the synthesis of carotenoids, antibiotics, and staphyloferrins, and only in the core genome is there the pathway for the synthesis and degradation of ketone bodies.

The only unique pathway in the core genome of *Enterobacter* spp. is the 00511 pathway of glycan degradation. And there are four unique pathways in the pangenome. These are pathways of dioxin and atrazine degradation, the linolenic acid metabolism pathway, and the 01054 pathway (non-ribosomal peptide structures). Similarly, with modules, the number of unique pathways in the core genome of *P. aeruginosa* significantly exceeds the number of those in the pangenome (two vs. 10). The organism is characterized by the unique pathways of styrene, toluene, chlorobenzene, and chlorocyclohexane degradation. In addition, it is noteworthy that only *P. aeruginosa* has non-homologous end joining (NHEJ) and biofilm formation pathways. The presence of a flagella assembly pathway in the core genome is also noteworthy.

*Mycoplasma* spp. do not have any unique pathways in either the pangenome or the core genome. Based on the results of the core genome comparison, all pathway annotations of *Mycoplasma* spp. are in the SEPM group, i.e., they are universal and characteristic for each strain of each investigated species. Additionally, *Mycoplasma* spp. share one annotation with *Enterobacter* spp. (glycan degradation) and one common annotation with *Enterobacter* spp. and *P. aeruginosa* (sphingolipid metabolism).

## 3. Materials and Methods

### 3.1. KEGG Databases

To compare the genomes of various organisms, the KEGG databases (https://www.genome.jp/kegg/) were used. KEGG is a collection of databases related to genomes, biological pathways, diseases, drugs, and chemicals. KEGG is used for bioinformatics research, including analysis of genomic, metagenomic, proteomic, and other data. In total, KEGG includes 11 databases, of which three were used in this study: KEGG ORTHOLOGY, KEGG MODULE, and KEGG PATHWAY. For comparison of single genomes, the KEGG 90.0 release was used, and for comparison of pangenomes, the KEGG 91.0 release was used.

KEGG ORTHOLOGY is a database of molecular functions presented in terms of functional orthologs. The functional ortholog is defined manually in the context of KEGG molecular networks, namely KEGG pathway maps (KEGG PATHWAY) and KEGG modules (KEGG MODULE). Each group of genes and proteins is assigned to a unique KO identifier in accordance with the experimentally determined functions and context (for example, K01647 is citrate synthase). Higher-order annotations—modules and pathways—are formed from the groups of such identifiers.

KEGG MODULE consists of KEGG modules, indicated by identifiers starting with M (for example, M00009 is the Krebs cycle module). Each module is a set of proteins, represented by the corresponding identifiers of KO. The presence of an annotated module in an organism fully guarantees the presence of the function it implements. Specific reactions in the module can be implemented by alternative enzymes, which is reflected in the module scheme. For each organism, there is a list of modules annotated for it.

KEGG PATHWAY is a pathway map database that brings together objects from various databases, including genes, proteins, RNA, small chemicals, glycans, and chemical reactions. The path may include or partially intersect with multiple modules. So, for example, the Krebs cycle path includes three modules: M00009 is the complete module of the Krebs cycle, as well as M00010 and M00011 modules, corresponding to parts of the M00009 module. A specific pathway can be annotated in an organism, even if some of its parts are missing. In this study, this feature allows one to take into account pathways, even if they are naturally incomplete (as in many highly specialized organisms), or some of their elements are present, but not characterized in a given organism.

### 3.2. Organisms of Interest

The study used genomic data from the KEGG databases for four organisms:*Staphylococcus aureus.* Reference strain NCTC 8325, 53 genomes of *S. aureu*s were used.*Enterobacter* species. Reference strain EcWSU1, 35 genomes of *Enterobacter* spp. were used.*Pseudomonas aeruginosa.* Reference strain ATCC 15692, 21 genomes of *P. aeruginos*a were used.*Mycoplasma* species. Reference strain ATCC 43663, 92 genomes of *Mycoplasma* spp. were used.

### 3.3. Algorithm of Comparison

For comparison of single strains for each of the reference strains, the lists of annotated modules and genomic pathways were obtained. On the basis of the obtained lists, a combined list of found annotations was compiled and each of its elements was checked for its presence in each of the strains. Thus, the annotations, depending on the presence/absence in each of the four sets of genomes, could belong to one of 15 theoretically possible intersection groups (the maximum number of combinations is 2^4^ − 1).

However, the comparison of genomes of single strains gives a somewhat biased idea of the occurrence of annotations in a group of organisms. The comparison of groups of genomes allows us not only to find out whether the module is universal for organisms of a certain taxon, but also to shed light on the entire repertoire of annotations encountered.

The sets of annotations for different genomes of the same species were used for four organisms. Based on this, we created lists of all possible annotations for organisms and frequencies of appearance of those annotations.

Annotations occurring in each genome (frequency equal to one) were assigned to the taxon’s core genome. Annotations occurring in at least one genome (frequency over zero), including all, were assigned to the taxon pangenome. Thus, for each type of microorganism, four lists of annotations were obtained: modules of the core genome, modules of the pangenome, pathways of the core genome, and pathways of the pangenome. For the obtained lists, combined lists of annotations were compiled and their occurrence in each of the taxa was determined according to a scheme similar to the comparison of single genomes.

All procedures were done in Python 3.7 with the re, request, pandas, and numpy libraries.

## 4. Conclusions

In this study, pangenome and comparative analyses of 201 genomes of *Staphylococcus aureus*, *Enterobacter* species, *Pseudomonas aeruginosa*, and *Mycoplasma* species were carried out. Unique and universal features found in the analysis of core genomes and pangenomes of the studied organisms are presented in Figure 8. Based on the obtained data, it was concluded that:(1)Each *S. aureus* strain has multidrug resistance systems, even if they are susceptible to methicillin. Twenty-five percent of the strains in the KEGG database are methicillin resistant, 85% of the strains are tetracycline resistant, 45% of the strains have the beta-lactam resistance module, and 21% of the genomes have the QacA efflux system multidrug resistance module.(2)*Enterobacter* spp. have a wide range of antibiotic resistance systems, however, there are strains without such systems at all. Seventy-six percent of the strains have at least one such system. Twenty-one percent of the *Enterobacter* spp. strains in the KEGG database are resistant to carbapenems. Seventy-four percent of the strains have the AcrEF-TolC efflux system module.(3)*P. aeruginosa* has a wide range of unique efflux systems. One hundred percent of the strains have at least two drug resistance systems, and 75% of the strains have seven. Resistance to carbapenems was found in 14% of the strains. The M00718 module of the MexAB-OprM efflux system, which confers multidrug resistance, and the M00745 module of the OprD repression system, conferring resistance to imipenem, were found in 20 of the 21 genomes.(4)*Mycoplasma* spp. do not have annotated antibiotic resistance or antimicrobial peptide resistance modules, although resistance has been proven for some *Mycoplasma* species. Determination of resistance mechanisms requires additional research.(5)Each of the organisms has a characteristic set of metabolic traits, whose contribution to drug resistance can be determined in future studies.

## Figures and Tables

**Figure 1 ijms-21-07839-f001:**
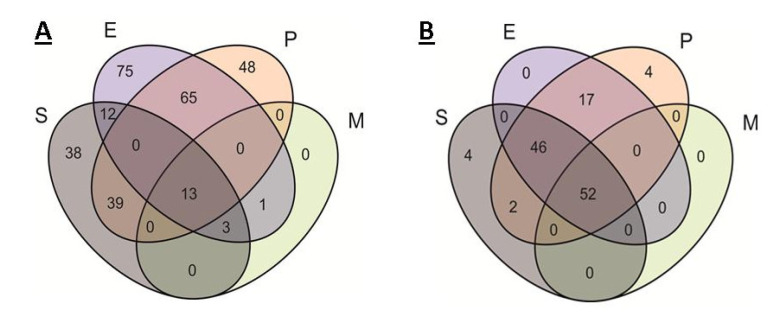
Venn diagram of the occurrence of different KEGG modules (**A**) and the presence of different pathways (**B**) in the organisms. S, E, P, and M indicate ellipses corresponding to *S. aureus*, *E. cloacae*, *P. aeruginosa*, and *M. mobile*.

**Figure 2 ijms-21-07839-f002:**
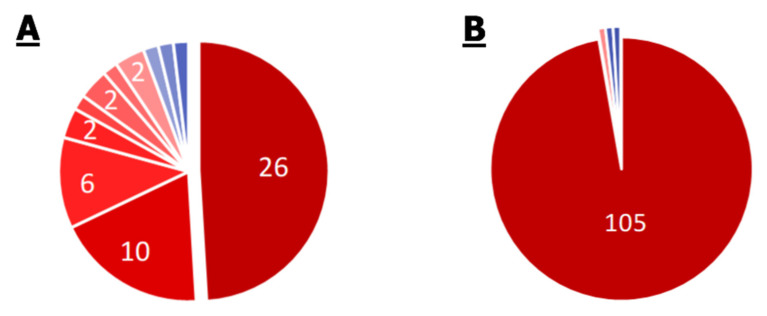
Pie chart of the proportion of annotations in the 53 genomes of *S. aureus* strains. Numbers depict annotations with equal frequency. Frequency—from 0.02 to 1—is presented in the color of slices on the scale from blue to red for distribution of modules (**A**) and pathways (**B**). Small slices without numbers correspond to the sole annotation with such frequency. Among 26 core modules (with frequency equal to 1), there are two modules of multidrug resistance, two modules of AMP resistance, and 22 different metabolic modules. Among 10 modules with frequency equal to 0.98, all modules are metabolic. Among six modules with frequency equal to 0.85, there are two of multidrug resistance and one of AMP resistance, the rest are metabolic. The beta-lactam resistance module is present in 45% of genomes, the methicillin resistance module in 24%, and multidrug resistance efflux pump QacA in 21%. The rest of modules are metabolic. Ninety-seven percent of pathways are present in every genome.

**Figure 3 ijms-21-07839-f003:**
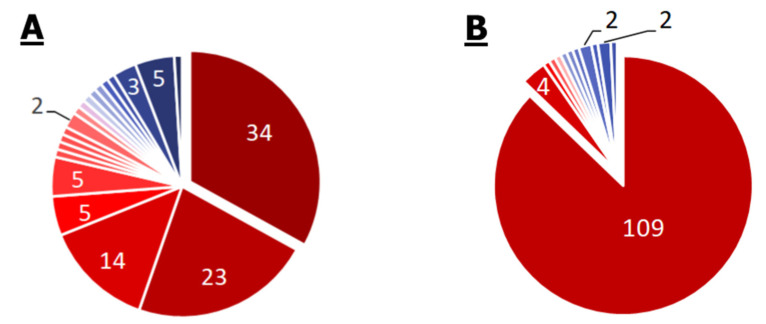
Pie chart of the proportion of annotations in the 35 genomes of *Enterobacter* spp. strains. Numbers depict annotations with equal frequency. Frequency—from 0.03 to 1—is presented in the color of slices on the scale from blue to red for distribution of modules (**A**) and pathways (**B**). Small slices without numbers correspond to the sole annotation with such frequency. There are no resistance modules among the 34 core modules, in 23 modules with frequency equal to 0.97, or in 14 modules with frequency equal to 0.94. They are only metabolic. Efflux pump AcrEF-TolC is present in 74% of genomes, the carbapenem resistance module in 31%, and the module of OmpF porin repression in 3% (only one strain). The rest of the modules are metabolic. Eighty-seven percent of pathways are present in every genome and 3% more are present in 97% of genomes.

**Figure 4 ijms-21-07839-f004:**
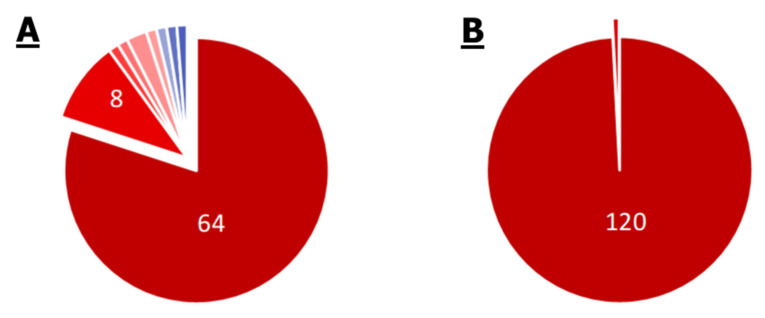
Pie chart of the proportion of annotations in the 21 genomes of *P. aeruginosa* strains. Numbers depict annotations with equal frequency. Frequency—from 0.15 to 1—is presented in the color of slices on the scale from blue to red for distribution of modules (**A**) and pathways (**B**). Small slices without numbers correspond to the sole annotation. Among 64 core modules, there are two multidrug resistance modules: efflux pumps MexCD-OprJ and MexJK-OprM. Among eight modules with frequency equal to 0.95, there are efflux pump MexAB-OprM and imipenem resistance modules, the rest are metabolic modules. Efflux pump MexEF-OprN and MexPQ-OpmE are present in 81% of genomes, efflux pump MexXY-OprM in 76%, and the carbapenem resistance module in 14%. The rest of the modules are metabolic. One hundred and twenty out of 121 pathways are present in every genome and one pathway has frequency of 0.95.

**Figure 5 ijms-21-07839-f005:**
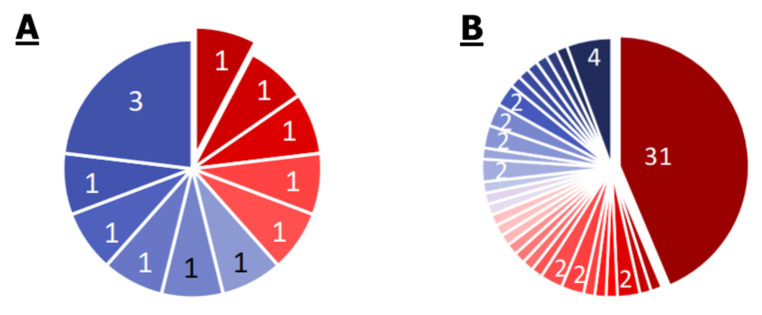
Pie chart of the proportion of annotations in the 92 genomes of *Mycoplasma* spp. strains. Numbers depict annotations with equal frequency. Frequency—from 0.01 to 1—is presented in the color of slices on the scale from blue to red for distribution of modules (**A**) and pathways (**B**). Small slices without numbers correspond to the sole annotation with such frequency. All the modules in the pangenome are metabolic and no modules of resistance are found. Distribution of pathway frequencies is remarkably dispersed. Only 42% of pathways are present in all genomes.

**Figure 6 ijms-21-07839-f006:**
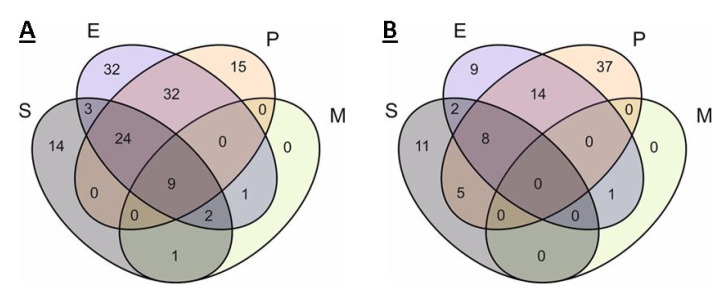
Venn diagram of the occurrence of different KEGG modules in the pangenome (**A**) and core genome (**B**) in organisms. S, E, P, and M indicate ellipses corresponding to *S. aureus*, *Enterobacter* spp., *P. aeruginosa*, and *Mycoplasma* spp.

**Figure 7 ijms-21-07839-f007:**
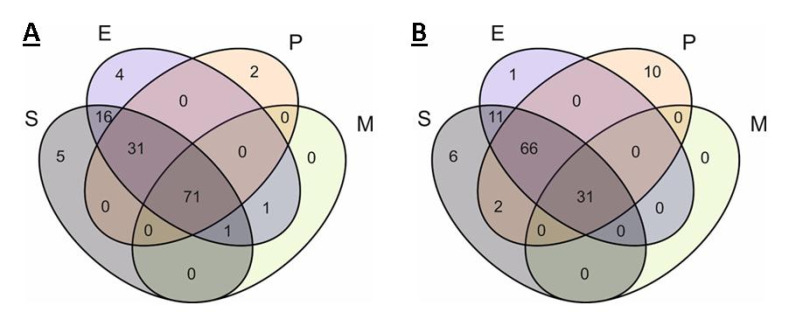
Venn diagram of the occurrence of different KEGG pathways in the pangenome (**A**) and core genome (**B**) in organisms. S, E, P, and M indicate ellipses corresponding to *S. aureus*, *Enterobacter* spp., *P. aeruginosa*, and *Mycoplasma* spp.

**Figure 8 ijms-21-07839-f008:**
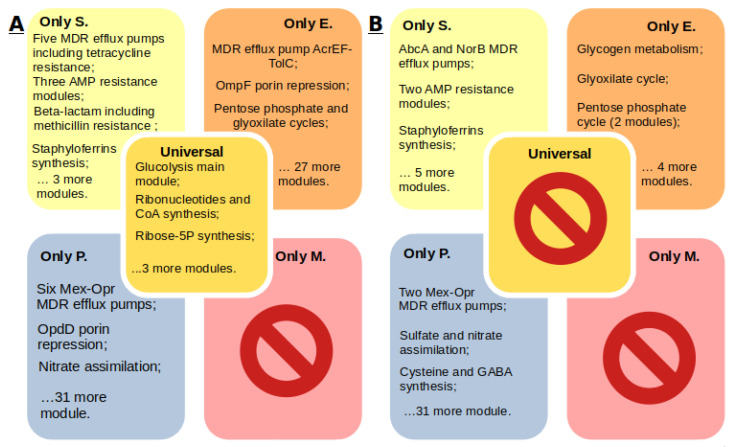
Unique and universal features found in the analysis of pangenomes (**A**) and core genomes (**B**) of *S. aureus* (S)*, Enterobacter* spp. (E), *P. aeruginos*a (P), and *Mycoplasma* spp. (M). MDR denotes multidrug resistance.

**Table 1 ijms-21-07839-t001:** Number of genes, modules, and pathways for the reference strains in the KEGG database.

	Protein Genes	RNA Genes	KEGG Modules	KEGG Pathways
*S. aureus*	2767	77	105	104
*E. cloacae*	4619	108	208	118
*P. aeruginosa*	5572	106	165	121
*M. mobile*	635	34	17	52

**Table 2 ijms-21-07839-t002:** Distribution of modules among categories and subcategories. S, E, P, and M indicate the presence of organisms in certain categories and represent *S. aureus*, *E. cloacae*, *P. aeruginosa*, and *M. mobile*, respectively.

	Total	Drug Resistance	Metabolic	Transport	Regulation	AMP Resistance
S	38	7	6	12	10	3
P	48	10	14	11	13	0
E	75	5	27	30	9	4
EP	65	5	24	26	10	1
SE	12	0	3	9	0	0
EM	1	0	1	0	0	0
SEP	39	0	26	11	2	0
SEM	3	0	1	2	0	0
SEPM	13	0	9	4	0	0

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
