# Peer review of "Comparative Analysis of Proteomes of a Number of Nosocomial Pathogens by KEGG Modules and KEGG Pathways"

_ijms, 2020, doi:10.3390/ijms21217839_

Round 1

Reviewer 1 Report

the work is conceptually interesting however the choice of pathogens analyzed seems not complete. it is well known from literature that The ESKAPE pathogens group (Enterococcus faecium, Staphylococcus aureus, Klebsiella pneumoniae, Acinetobacter baumannii, Pseudomonas aeruginosa, and Enterobacter species) are the leading cause of nosocomial infections throughout the world. It would therefore be interesting to extend the functional analysis to the pathogens included in the ESKAPE group.

Moreover, the graphic part should be integrated with some figures (e.g heatmaps or similar) that highlight the most important similarities and differences at the core and pangenome level, as well as the unique metabolic pathways. it should not be limited to quantitatively indicate the differences but also qualitatively (e.g. the name of a specific module) to provide a visual feedback easily usable by the reader of the differences or similarities detected in the analysis.

Author Response

Comments and Suggestions for Authors

the work is conceptually interesting however the choice of pathogens analyzed seems not complete. it is well known from literature that The ESKAPE pathogens group (Enterococcus faecium, Staphylococcus aureus, Klebsiella pneumoniae, Acinetobacter baumannii, Pseudomonas aeruginosa, and Enterobacter species) are the leading cause of nosocomial infections throughout the world. It would therefore be interesting to extend the functional analysis to the pathogens included in the ESKAPE group.

Response 1: Thanks to the reviewer for his valuable suggestions. It would be very interesting to consider this group of pathogens in our work. We are currently developing AMPs and testing them for specially selected organisms. Therefore, the choice of pathogens was dictated by our antibacterial peptide developers.

Moreover, the graphic part should be integrated with some figures (e.g heatmaps or similar) that highlight the most important similarities and differences at the core and pangenome level, as well as the unique metabolic pathways. it should not be limited to quantitatively indicate the differences but also qualitatively (e.g. the name of a specific module) to provide a visual feedback easily usable by the reader of the differences or similarities detected in the analysis.

Response 2: We have redrawn Figures 2-5 according to the reviewer’s comments and have added one new summarizing Figure of comparative analysis.

Reviewer 2 Report

In this study, Slizen and Galzitskaya presented an interesting work  describing pangenome and comparative analysis of 201 genomes of Staphylococcus aureus, Enterobacter species, Pseudomonas aeruginosa and Mycoplasma species using high-level functional annotations — KEGG pathways and KEGG modules. In my opinion, the study design is appropriate, the methodologies applied are sound, the results are clearly presented and the conclusion is fully supported by the results. However, there are still some minor points that need to be described before this manuscript can be accepted for publication in IJMS.

  1. In the Results and discussion section, line 119, it should be “ antimicrobial peptide (AMP) resistance”.
  2. In Table 2, line 129, there are some words missing in the column names: “Drug re-”, “Transporta-”, “Regula-”
  3. Although the authors tried to compare the genomes of nosocomial pathogens with each other and with Mycoplasma genomes in the Results and discussion section, but what I mainly see are only the results of the KEGG pathways and KEGG modules. The authors should describe in more detail about the versatility of certain metabolic features and mechanisms of drug resistance regarding the differences between the nosocomial pathogens and the Mycoplasma spp. in order to increase the novelty and significance of the current study.

Author Response

Comments and Suggestions for Authors

 In this study, Slizen and Galzitskaya presented an interesting work describing pangenome and comparative analysis of 201 genomes of Staphylococcus aureus, Enterobacter species, Pseudomonas aeruginosa and Mycoplasma species using high-level functional annotations — KEGG pathways and KEGG modules. In my opinion, the study design is appropriate, the methodologies applied are sound, the results are clearly presented and the conclusion is fully supported by the results. However, there are still some minor points that need to be described before this manuscript can be accepted for publication in IJMS.

  1. In the Results and discussion section, line 119, it should be “ antimicrobial peptide (AMP) resistance”.
  2. In Table 2, line 129, there are some words missing in the column names: “Drug re-”, “Transporta-”, “Regula-”
  3. Although the authors tried to compare the genomes of nosocomial pathogens with each other and with Mycoplasma genomes in the Results and discussion section, but what I mainly see are only the results of the KEGG pathways and KEGG modules. The authors should describe in more detail about the versatility of certain metabolic features and mechanisms of drug resistance regarding the differences between the nosocomial pathogens and the Mycoplasma spp. in order to increase the novelty and significance of the current study.

Response 1: We would like to thank the reviewer for the valuable suggestions. We have corrected all mentioned typos and errors. We have added more details in the text and new Figure 8 summarizing our comparative analysis.

Round 2

Reviewer 1 Report

the paper could be accepted in present form